# Hyperoxidation of Peroxiredoxins and Effects on Physiology of *Drosophila*

**DOI:** 10.3390/antiox10040606

**Published:** 2021-04-15

**Authors:** Austin McGinnis, Vladimir I. Klichko, William C. Orr, Svetlana N. Radyuk

**Affiliations:** Department of Biological Sciences, Southern Methodist University, Dallas, TX 75275, USA; amcginnis@mail.smu.edu (A.M.); vklichko@smu.edu (V.I.K.); borr@smu.edu (W.C.O.)

**Keywords:** peroxiredoxin, sulfiredoxin, redox state, hyperoxidation, *Drosophila*

## Abstract

The catalytic activity of peroxiredoxins (Prx) is determined by the conserved peroxidatic cysteine (Cys_P_), which reacts with peroxides to form sulfenic acid (Cys-SOH). Under conditions of oxidative stress, Cys_P_ is oxidized to catalytically inactive sulfinic (Cys-SO_2_) and sulfonic (Cys-SO_3_) forms. The Cys-SO_2_ form can be reduced in a reaction catalyzed by sulfiredoxin (Srx). To explore the physiological significance of peroxiredoxin overoxidation, we investigated daily variations in the oxidation state of 2-Cys peroxiredoxins in flies of different ages, or under conditions when the pro-oxidative load is high. We found no statistically significant changes in the 2-Cys Prxs monomer:dimer ratio, which indirectly reflects changes in the Prx catalytic activity. However, we found daily variations in Prx-SO_2/3_ that were more pronounced in older flies as well as in flies lacking Srx. Unexpectedly, the *srx* mutant flies did not exhibit a diminished survivorship under normal or oxidative stress conditions. Moreover, the *srx* mutant was characterized by a higher physiological activity. In conclusion, catalytically inactive forms of Prx-SO_2/3_ serve not only as a marker of cellular oxidative burden, but may also play a role in an adaptive response, leading to a positive effect on the physiology of *Drosophila melanogaster*.

## 1. Introduction

The peroxiredoxin (Prx) protein family plays an important role in maintaining redox homeostasis and is functionally conserved in all kingdoms of life. Besides functional conservation, there is substantial sequence conservation of peroxiredoxins [1]. The most highly conserved region surrounds a key redox-active cysteine, termed the peroxidatic cysteine, Cys_P_, which is reversibly oxidized by peroxides to a sulfenic acid (Cys-SOH). This peroxidatic Cys is present in all Prx subgroups, 1-Cys (Prx6), 2-Cys (Prxs 1–4) and atypical 2-Cys Prx (Prx5). In the case of the 2-Cys subgroup, Cys-SOH reacts with an additional conserved Cys residue termed the resolving cysteine, Cys_R_, to form intra- or intermolecular disulfide bonds, that are in turn resolved through interaction with a variety of thiols to regenerate the fully reduced Prx. Under conditions of excess reactive oxygen species (ROS), the key cysteine can be selectively overoxidized to catalytically inactive sulfinic (Cys-SO_2_) and sulfonic (Cys-SO_3_) forms [2,3].

Peroxiredoxins vary in their susceptibility to overoxidation due differences in their structure, physicochemical characteristics, subcellular localization and availability of reducing substrates, as well as the kinetics of dimerization of the sulfenic acid intermediate [4]. Studies in mammals have shown that 2-Cys Prxs belonging to type Prx1, Prx2 and Prx3 are most vulnerable to hyperoxidation [5,6,7,8], while 2-Cys Prx4 and atypical 2-Cys Prx5 are resistant [9,10]. Although the conditions that lead to Prx overoxidation are well established, the physiological significance of Prx overoxidation remains underexplored.

The reduction of hyperoxidized peroxiredoxin to its peroxidatically active form is catalyzed by sulfiredoxin (Srx) [11]. To shed more light on the in vivo functions of overoxidized Prxs, we investigated the oxidation state variations of the *Drosophla melanogaster* Prxs in the presence or absence of sulfiredoxin. In particular, we tracked the oxidation state of 2-Cys Prxs in flies as a function of age and the diurnal cycle and variations in the accumulation of *Drosophila* Prx forms oxidized to different degree and also investigated the phenotype of the *srx* mutant. We also investigated the oxidation state of *Drosophila* 2-Cys Prxs in flies of different ages, as well as diurnal variations, to determine the effects of Prx oxidation on physiology and aging. We also determined a role of Srx, an important factor in Prx recycling, in fly physiology and aging.

## 2. Materials and Methods

### 2.1. Fly Strains and Procedures

To exclude background effects, all mutant, driver and transgenic fly lines were backcrossed to our *yw* control strain a minimum of 8 times in order to obtain genetically homogeneous stocks. The daughterless Da-GAL4 driver line was supplied by Dr. Blanka Rogina (University of Connecticut Health Science Center). Under- and overexpression of dPrx3 was achieved using UAS-RNAi-*dprx3* and UAS-dPrx3 transgenic fly lines described in [12]. Under- and overexpression of dPrx4 was achieved using UAS-RNAi-*dprx4* and UAS-dPrx4 transgenic fly lines described in [13]. For under-expression of *Drosophila* homolog of mammalian Prx1 or Prx2, also known as CG1633, *jafrac1*, DPx-4783 and DmTPx-1, the UAS-RNAi construct KK110159 received from Vienna Drosophila Resource Center (stock # v109514) was used. Jafrac1 is more closely related to the mammalian Prx2, based on the features of the amino acid sequence [14], thus it is named here and thereafter dPrx2. For overexpression of another *Drosophila* homolog of Prx1/Prx2, also known as CG6888, UAS-RNAi constructs KK105974 and GD10756 received from Vienna Drosophila Resource Center (stock # v26094 and v107556) were used. Although CG6888 is homologous to both mammalian Prx1 and Prx2, to distinguish it from dPrx2, it is named here dPrx1. To overexpress dPrx1, UAS-dPrx1 transgenic lines were generated by cloning the entire coding region of the CG6888 gene into the pUAST vector. The recombinant pUAS-dPrx1 plasmid DNA was sent to the TheBestGene Co. (http://www.thebestgene.com, accessed on 14 April 2021) for P-element transformation. Two different transgenic lines (UAS-dPrx1–1 and −2) carrying the UAS-dPrx1 construct were selected and used in the experiments. A strain containing a P-element insertion 244 bp downstream of the start codon of the CG6762 gene (*Drosophila* homolog of mammalian Srx), CG6762^G1102^ (stock # 33537), was obtained from the Bloomington Stock Center.

Flies were maintained on standard sucrose–cornmeal fly food at 25 °C on a 12 h light/dark cycle. For experiments, flies were collected within 1–2 days after hatching, followed by separation of males and females. Approximately 25 flies were placed in each vial. Survivorship studies were conducted as described in our previous publications [12,13]. Oxidative stress was elicited by feeding flies with a 1% sucrose solution containing 0.5 M H_2_O_2_.

### 2.2. Immunoblot Analysis

Immunoblot analyses were essentially performed as described previously [15]. Briefly, proteins for immunoblot analysis under reducing conditions were extracted with T-PER tissue protein extraction reagent (Pierce) containing protease inhibitors (Roche). For non-reducing conditions, extracts were made using a non-reducing lysis buffer (50 mM Tris pH 8.0, 150 mM NaCl, 0.1% SDS, and 0.5% NaDeoxylate) that contained a protease inhibitor cocktail (Roche). To trap Prxs in their native redox state and prevent over-oxidation, protein extracts were made with a non-reducing lysis buffer containing 100 mM N-ethyl maleimide (NEM). Antibodies that recognize *Drosophila* dPrx2, dPrx3 and dPrx4 are described in our publications [12,13,16]. Anti-actin antibodies (MP Biomedicals) were used as a control for loading. The antibodies (anti-Prx-SO_2/3_ antibody) generated against the conserved peptide sequence containing a SO_3_-cysteine were purchased from Abcam. Anti-dPrx1 antibodies were raised against purified recombinant dPrx1 protein, produced in an *E. coli* expression system, using the services of Covance Research Products, Inc. Images were obtained using a ChemiDocTM Touch Imaging system (Bio-Rad, Hercules, CA, USA) and analyzed using ImageLab software v.5.2.1 (Bio-Rad, Hercules, CA, USA).

### 2.3. Locomotor Activity Analysis

A TriKinetics Locomotor Activity Monitoring System (TriKinetics Inc., Waltham, MA, USA), Data Acquisition (DAMSystem308X) and File Scan (DAMFileScan110X) Software were used to measure locomotor activity. Studies were conducted under a 24 h light/dark cycle regimen (12 h:12 h LD) at 25 °C and 50% humidity. Data was recorded as number of crossings per 10 min bin. Fly activity data were analyzed using R version 3.1.2, RStudio Version 0.98.1091, Microsoft Excel Version 14.6.1 and Prism 5.0c (GraphPad Software, San Diego, CA, USA) to calculate total activity over time. An R script was written by Olena Odnokoz and used to organize the data in the format represented in the figures (script available upon request).

### 2.4. Statistical Methods

Data were statistically analyzed using GraphPad Prism 5.0c and Microsoft Excel. Statistical significance of differences between survival curves was assessed using the log rank test. Differences in the Prx protein levels and the levels of overoxidized Prxs were assessed by paired Student’s *t*-tests using Microsoft Excel software.

## 3. Results

There are four mammalian 2-Cys peroxiredoxin isoforms that form intermolecular disulfide bonds during their catalytic cycle, Prxs 1, 2 3 and 4 [17]. All four have homologs in *Drosophila* [18]. Homologues of mammalian Prx1 and 2 are represented by two genes, CG1633 (also known as jafrac1, DPx4783, dPrx-1) and CG6888. Structurally, CG1633 has 73% and 71% identity with human PRDX1 and PRDX2, while CG6888 is considered more orthologous to Prx2 (60% and 57% identity to human PRDX2 and PRDX1, FlyBase data). These *Drosophila* paralogs share 61% amino acid identity and are co-localized to the cytosol ([18], FlyBase and unpublished observations). Like its mammalian counterpart, the *Drosophila* equivalent of Prx3 (dPrx3 or DPx5037) is localized to the mitochondria [18]. Finally, *Drosophila* also possesses a Prx4 homolog, designated as jafrac2, DPx4156 or dPrx4, and, like its mammalian ortholog, is found in the endoplasmic reticulum (ER), and a variety of other locations, including the extracellular space and cytosol [13,18].

### 3.1. Ratios of Prx Dimer:Monomer as an Indirect Indicator of Their Activity

Peroxidase activity of Prx depends on the oxidation of a conserved cysteine residue at the active site and formation of different modifications, including intra- and intermolecular disulfide bonds. The disulfides are ultimately reduced by thioredoxin (Trx) and/or glutathione with glutathione S-transferase (GSH + Gst), thereby bringing the functional cysteines to catalytically active reduced thiol states (reviewed in [19,20,21]). During this process, Prx subunits alternate between monomeric and covalently linked dimeric forms that migrate differently on non-reducing gels. Thus, one of the features that may reflect activity of Prxs is the dimer/monomer ratio. The dimeric form is the least reactive form of Prxs and is well protected from overoxidation. Some Prxs (Prx3, Prx2) are mainly found as dimers, which are rapidly reduced to monomers under the conditions of oxidative stress and react with peroxides as they form [4,22]. Thus, the Prx dimer/monomer ratio may reflect not only the susceptibility of different Prxs to oxidation but also changes in peroxide levels that fluctuate daily [23] and also accumulate during aging [24,25]. Consequently, we have investigated the daily variations in the proportions of monomers and dimers for each individual 2-Cys *Drosophila* Prx (dPrx) in the heads and bodies of young (~5 da) and old flies (~50–55 da) (Figure 1).

Under standard conditions, dPrx4 was found to exist exclusively as a dimer in both young and old flies (Appendix A). We also observed no changes in the dPrx4 protein levels during the day and only a slight decrease in old flies, consistent with previous findings [13]. In the case of dPrx3, there was a measurable monomeric component, but the dimeric form was found to predominate. No significant daily variations were determined. We also observed diurnal fluctuations in the dimer:monomer ratio for the *Drosophila* Prx2 homolog in the heads of young and old flies, although not statistically significant. We were not able to reliably evaluate the dimer/monomer ratio of dPrx1 in heads, since expression levels of this protein in the *Drosophila* heads are very low (data not shown). However, we determined significant changes in the dimer:monomer ratio in the bodies of dPrx1 and a statistically significant increase in the overall ratio of dimers/monomers between young and old flies (Figure 1B), suggesting that concentrations of catalytically active dPrx change during aging, thus indirectly indicating the overall pro-oxidative changes in the state of catalytically active sulfhydryls.

### 3.2. Analysis of Prx Hyperoxidation

Some reports show that the state of oxidation of a catalytic cysteine of Prxs in different cells and organisms, including *Drosophila*, oscillates between reduced and oxidized forms [26]. It was suggested that, thereby, these switches reflect changes in the hydrogen peroxide metabolism. In assays aimed at the evaluation of overoxidized peroxiredoxin, antibodies generated against the conserved peptide sequence containing a SO_3_-cysteine are used [27]. Such antibodies were used in studies of circadian oscillations of PrxSO_2/3_ in many different species, from cyanobacteria to humans [26,28]. Since the catalytic site is highly conserved in 2-Cys *Drosophila* Prxs [18], all four Prxs can contribute to the signal obtained with antibodies raised against the hyperoxidized sulfonated peptide DFTFVC*PTEI.

Initially, we thoroughly evaluated the specificity of Prx-SO_2/3_ bands in *Drosophila* by using antibodies generated in our laboratory that recognize individual *Drosophila* Prx proteins (anti-dPrx1–4 and unpublished) [12,13,18]. We combined this with genetic manipulation of the *Drosophila* dPrx genes. The latter exploited the Gal4/UAS and RNAi systems to overexpress and knock down each dPrx; extracts were prepared from dPrx1–4 overexpression and RNAi strains as well as a control (*yw*) strain (Figure 2). We first probed immunoblots of the different extracts with the anti-Prx-SO_2/3_ antibody. We then stripped and re-probed the blots with the anti-dPrx1, 2, 3 and 4 antibody, which unambiguously identified the Prx dimer and monomer bands (Figure 2). In the case of dPrx 1–3, the bands migrated at their predicted molecular weights (44 kDa for a dimer and 22 kDa for a monomer, respectively) (Figure 2A–C), while the dPrx4 monomer and dimer migrated as 25 and 50 kDa proteins (Figure 2D). There was no SO_2/3_ signal at the position of dPrx4 monomer and the ~50 kDa band recognized by SO_2/3_ antibodies that corresponded by molecular weight (MW) to the dPrx4 dimer was observed in mutants and overexpressors, suggesting that this signal is non-specific.

Re-probing of the blots with the dPrx-specific antibody revealed that the ~22 kDa signals obtained with the anti-Prx-SO_2/3_ antibody co-migrated with the signals belonging the bona fide 22 kDa monomers of dPrx1–3 proteins (Figure 2A–C). This suggests that the Prx-SO_2/3_ antibodies likely recognize overoxidized ~22 kDa peroxiredoxin monomer, which can belong to dPrx1, 2 and 3 since these dPrxs are indistinguishable by their MW.

We then analyzed the temporal changes in dPrxs’ hyperoxidation. Further analysis of hyperoxidized Prxs relied on identification of ~22 kDa bands, whose specificity has been confirmed by re-probing of the blots with dPrx-specific antibodies (dPrx1, dPrx2 or dPrx3). Since much of the impactful gene regulation for the circadian system occurs in the brain, we initiated the analysis of dPrx overoxidation in the material prepared from the heads of control (Canton) flies. Only weak evidence for dPrx hyperoxidation was determined in the fly heads (Appendix A). However, there was a prominent band recognized by SO_2/3_ antibodies, which overlapped with signals obtained with dPrx1–3 proteins (representative immunoblot images are shown in Appendix A). To address potential age-related variations, the analysis was conducted in samples prepared from young (5 da) and old (50 da) flies and showed somewhat higher levels of overoxidized Prx in older flies, with a peak at 12–20 h zeitgeber time (ZT), or at the beginning and in the middle of the dark period (Figure 3).

### 3.3. The Effects of Prx1 and Prx2 Underexpression

The observed subtle difference in the proportion of dimers/monomers, as well as hyperoxidized dPrx forms between young and old flies, suggests a decrease in the catalytic activity of Prxs during aging. This decrease may be partly due to changes in Prx protein levels. Previously, we did not find noticeable changes in age-related expression of the dPrx3 protein [12], but we found an approximately 50% decrease in dPrx4 protein during aging [13]. Here, we examined the expression levels of the dPrx1 and dPrx2 proteins and did not find significant age-related changes (data not shown).

We also believe that the decrease in Prx activity due to oxidation-related inactivation may, at least in part, correlate with the effects of Prx underexpression. Since *Drosophila* Prx4 is not prone to overoxidation (Figure 2 and Appendix A), we focused on the effects of underexpression of Prxs 1–3. As previously reported, underexpression of dPrx3 did not lead to a pronounced phenotype and was characterized by only a slight decrease in resistance to oxidative stress (OS) [12].

Here, we investigated the effect of underexpression of dPrx1 and dPrx2 on survival under normal and oxidative stress conditions. Downregulation of dPrx2 using corresponding RNAi transgenes (see Materials and Methods) and high-level global drivers (DA-GAL4) resulted in a lethal phenotype with a small number of short-lived adult progeny (data not shown). In contrast to dPrx2, underexpression of the nearest dPrx2 paralog, dPrx1 (CG6888), does not result in any discernible phenotypes (Appendix A and Table 1). Thus, underexpression of Prxs had differential effects on the survival of flies under normal and OS conditions.

### 3.4. Hyperoxidation of Prxs in the Srx Mutant

While functional Prx peroxidatic cysteines are recycled from disulfides to reduced sulfhydryls during the catalytic cycle involving Trx/GSH systems, overoxidized cysteines are reduced by Srx, which replenishes the catalytically active forms of thiols. Therefore, Srx mutants are characterized by an increased level of hyperoxidized forms of Prx [29]. However, the functional significance of this event and the exact role of Srx in physiology have not yet been fully understood.

Here, we have characterized a *Drosophila* mutant for the gene CG6762 believed to be an ortholog to the mammalian Srxs [30]. This *Drosophila* gene shares high structural amino acid identity with mammalian Srxs and has a conserved Cys123 residue in the active site of Srxs (Cys99 in human Srx), as well as a conserved Srx motif SFGGCHR, surrounding this conserved Cys residue (bold, underlined) (Appendix A), which is responsible for reacting with peroxiredoxin Cys_P_ and formation of thiosulfinate intermediate (Prx-S_P_O-S-Srx). It was predicted that this *Drosophila* gene has sulfiredoxin activity.

In our experiments, we used the allele containing P-element insertion in the coding region of the CG6762 gene (CG6762^G1102^) 124 nucleotides downstream from the start codon. RT-PCR analysis revealed that flies homozygous for the mutant *srx* allele (*srx*−/−) had no detectable Srx product, as was determined by RT-PCR (Appendix A).

We evaluated the levels of the Prx-SO_2/3_ forms in the putative *Drosophila srx* mutant, which, similarly to the data obtained on *srx* mutants of other organisms, were characterized by the accumulation of overoxidized Prx forms (Figure 4 and Appendix A). From this observation, we inferred that this protein does indeed possess sulfinic acid reductase activity and appears to be a true homolog of mammalian and yeast Srx [11,29,31].

We also observed diurnal changes in the intensity of signals corresponding to the hyperoxidized forms of Prxs, with the highest levels observed at 12 ZT or at the end of the 12 h light period, when flies normally demonstrate higher activity (Figure 4). This increase also likely reflects an increase in H_2_O_2_ levels, for which fluctuations display similar profiling [23]. This is also consistent with the data obtained by the Sue Goo Rhee group [32], where higher levels of Prx-SO_2/3_ were observed in the *srx* mouse mutant, peaking around ZT12.

### 3.5. The Effect of Sulfiredoxin Underexpression on the Physiology of Drosophila

The data obtained with cultured mammalian cells and yeast indicate that Srx underexpression generally results in increased susceptibility to oxidative stress, whereas enhanced expression has a protective effect [33]. The antioxidant function of Srx has also been demonstrated in mice [34,35]. An unexpected finding of our study was that the *srx* mutant, despite a significantly higher degree of Prx hyperoxidation, had comparable to control or even slightly increased resistance to exogenous oxidative stress caused by H_2_O_2_ (Figure 5A). There was also no negative effect of Srx underexpression on lifespan (Figure 5B). Thus, this underexpression of Srx produces a phenotype that is almost indistinguishable from control in terms of survival under normal or oxidative stress conditions. Another surprising finding from the study is that levels of physical activity were higher in flies underexpressing Srx, as we found a significant increase in the locomotor activity in the *srx* mutants (Figure 6 and Appendix A).

## 4. Discussion

The main finding of the study is that the hyperoxidation of Prxs observed in the *srx* mutant does not adversely affect resistance to oxidative stress or lifespan of flies, but even leads to increased physical fitness and endurance. Srx is an enzyme whose primary biochemical function is to reduce hyperoxidized typical 2-Cys-containing Prxs [27,36,37].

There are numerous studies demonstrating the important role of Srx as an antioxidant protein that prevents oxidative stress damage to cells and confers increased resistance to oxidative stress by reducing hyperoxidized 2-Cys peroxiredoxins. Thus, in numerous studies on cell cultures [33], mammals [38] and plants [39], it was shown that underexpression of Srx can reduce the Prx activity due to accumulation of catalytically inactive Prx-SO_2/3_ forms, which leads to increased apoptosis and damage to various tissues, including neuronal tissue [40]. For instance, by affecting the activity of mitochondrial Prx3, Srx can influence the release of mitochondrial H_2_O_2_ [41]. In contrast, overexpression of Srx prevents many oxidative damages and apoptosis by modulating the activity of the stress response, inflammatory and apoptotic pathways [33]. In this way, Srx may confer a neuroprotective function [42,43,44] or rescue cardio tissues from injury [45] or play a role in kidney protection [46]. Srx is also essential for normal responses to receptor stimulation in endocrine system signaling [47].

On the other hand, by reducing ROS and affecting apoptosis, Srx may favor the survival of cancerous cells and thereby contribute to carcinogenesis [48]. For instance, Srx underexpression can sensitize cancer cells to ER stress-induced cell death [49], increase apoptosis of tumor cells and thus decrease tumor cell proliferation, while the opposite effects have been observed in Srx overexpressors, which correlated with more aggressive cancers and poor prognosis [8,48,50,51,52,53].

Thus, by regulating Prx oxidation and the release of ROS, and thereby influencing the inflammatory, apoptotic and stress response pathways, Srx can have both positive and negative effects on physiology and lifespan. However, the positive effect on the physiology of *Drosophila* found in our studies suggests that the effects of Srx underexpression are unlikely to be associated with suppression of abnormal cell proliferation and carcinogenesis. We did not use the *Drosophila* cancer model, where Srx can interfere with the pathways leading to cancer development. Despite the established properties of Srx to promote the survival of both cancer and other cells, including neuronal ones, the discovered positive effects of the *srx* mutant on the survival of *Drosophila* under OS conditions caused by H_2_O_2_ consumption rather suggest adaptation to pro-oxidants, or the hormetic effect.

One plausible explanation is that the beneficial effects of Prx overoxidation in *srx* mutants may be associated with a gain in chaperone function, promoted by Prx hyperoxidation [31,36,54]. On the other hand, this may be due to redox signal build-up [55] and effects on signal transduction, or preserving the function of the Prx reducing equivalent, Trx [56]. For example, repetitive oxidative stress has been reported to lead to adaptation to pro-oxidants and enhancement of the Nrf2-independent antioxidant response [57], including upregulation of Srx. Srx itself is a target of the Nrf2 pathway [58] and is activated during the first exposure to oxidants via Nrf2 signaling. The Nrf2-mediated antioxidant response is expected to be weaker after repeated exposure to stress, since sufficient protection has already been established. During the second treatment with oxidants, the Nrf2 activation reaches its limits; nevertheless, Srx activation continues to increase, indicating that the Srx response is probably not driven by Nrf2 alone, or that enhanced Srx activity suppresses Nrf2 activation due to antioxidant feedback effects. It is possible that Srx underexpression can, in turn, create a pro-oxidative environment and continuing oxidative stress, resulting in upregulation of Nrf2 signaling and enhancement in defense responses, which was observed in our experiments.

It is also possible that hyperoxidized Prxs in the *srx* mutant could signal as damage-associated molecular patterns (DAMPs) or alter post-translation modifications of other proteins, which in turn could signal as DAMPs and induce different response pathways. Thus far, we could not find any reports that hyperoxidized Prxs can serve as DAMPs, but there are reports that Prxs can modify other proteins by targeting protein sulfhydryls and altering the structure of target proteins, thereby causing them to act as DAMPs [59]. Thus, the peroxiredoxin-1 (Prdx1) released into the system and circulating is reported to act as a novel DAMP and induce a pro-inflammatory response upon liver injury [60]. Although the report did not indicate which forms of Prx might function, the *prx1-/-* mutant mice had increased protection from liver injury and inflammation. Likewise, Prx2 acts as a DAMP in ischemic stroke when released into the extracellular space [61]. Again, there has been no investigation on which form of Prx is responsible for the observed effects. Since the *prx1-/-* mouse mutant exhibited anti-inflammatory effects, it is possible then that the positive physiological effects observed in our study in the *Drosophila* srx mutant were associated with significant Prx inactivation and decreased inflammation, although further studies are needed to support this assumption.

Finally, other functions of Srx, such as denitrosylase activity [62] or degluathionylative activity [63], may be responsible for the observed phenotype of the *srx* mutant. It can also be a combination of different Srx functions with multiple targets [64]. However, nitrosylation of Prx was not investigated in this study; thus, this possibility remains speculative. Taken together, our data suggest that the observed hyperoxidation of Prxs in the *srx* mutant may promote an adaptive response to continuing but moderate endogenous oxidative stress. It is possible that this adaptation is also responsible for the enhanced locomotor abilities and negative geotaxis of the *srx* mutant (Figure 6).

Another important goal of this study was to carefully investigate the oxidation state of Prxs during the day and during aging. The use of Prx mutants and overexpressors and Prx-specific antibodies helped to unambiguously identify signals belonging to Prx dimers and monomers. This, in turn, helped to validate the signals belonging to the hyperoxidized Prxs and provide reliable data pertaining to daily changes in Prx hyperoxidation in *Drosophila*. Thus, we were unable to observe circadian oscillations of the Prx-SO_2/3_ forms in the heads of flies, and the signals obtained with antibodies recognizing the hyperoxidized forms of Prxs were weak (Appendix A). This is not surprising, since, according to data obtained on other organisms, overoxidized peroxiredoxins accumulate only in cells with a higher level of oxidative stress [65,66,67]. In addition, according to some reports, the proportion of hyperoxidized Prxs is not significant and accounts for about 1–2% of the total Prx protein, as was observed with Prx1 [5].

We also determined that the overoxidized Prxs were found only at the ~22 kDa position, which corresponds to the molecular weight of the monomer for Prx1, Prx2 or Prx3, but were never present in signals corresponding to the molecular weight of the dimers (Figure 2). This explains the discrepancy with the results obtained by Edgar et al. [26], because their analysis, where they found circadian oscillations of hyperoxidized Prxs in the material extracted from *Drosophila* heads, was based on the identification of a putative peroxiredoxin dimer with a molecular weight of ~50 kDa, but this was not rigorously addressed.

Usually, Prx monomers are practically undetectable under normal conditions and dimers, in which Prx is protected from overoxidation, predominate. Thus, the proportion of Prx monomers that are prone to peroxidation during the catalytic cycle, when they are in the reduced form (CysP-SH), is usually relatively small. This is well established in mammals, in which mitochondrial Prx3 and cytosol-localized Prx2 are mainly detected as dimers, while Prx1 is present in both reduced (Prx_P_-SH) and oxidized (Prx_P_-S-S-Prx_R_) forms [6,22,65]. Our data obtained on *Drosophila* are consistent with these findings in mammals, since variants of the cytosolic dPrx, homologous to mammalian Prx1/2 and designated as dPrx1, mitochondrial dPrx3 and ER-localized dPrx4, are found as dimers (Figure 1 and Appendix A). An exception is the cytosolic Prx, designated in our studies as dPrx2, in which the dimer-to-monomer signal ratio varied within 1.5–5 (Figure 1).

However, other complexities should also be borne in mind, since Prx catalysis is determined by the dynamics of resolution of the transient form of oxidized peroxidatic cysteine, sulfenic acid. The latter, in turn, depends on many factors, including the availability of reducing equivalents and the concentration of peroxides. Nevertheless, the “static” dimer/monomer ratio model reflects to some extent the ability of Prxs to perform a catalytic cycle and allows the evaluation of the Prx peroxidatic activity.

In contrast to the data obtained with the fly heads (Appendix A), we were able to detect some diurnal changes in Prx-SO_2/3_ in the fly bodies, which were more prominent in old flies (Figure 3 and Appendix A), and these changes likely reflect changes in H_2_O_2_ concentrations known to vary throughout the day [23] and also increase with age [25]. The levels of Prx-SO_2/3_ were also significantly higher in the *srx* mutant and showed more pronounced variations with a higher magnitude (Figure 4 and Appendix A), peaking at ~12 h, which usually corresponds to the time of accumulation of oxidative damage [8]. This is consistent with the findings of the Rhee group that strong SO_2/3_ rhythms persist in the cells from Srx-deficient mice, presumably reflecting H_2_O_2_ levels [32]. However, this is not related to diurnal variations in Srx, which affect Prx overoxidation and which, as was found in a study [41], undergo anti-phasic circadian oscillation with mitochondrial Prx. Thus, the data indicate that the observed changes in Prx oxidation are unlikely to be associated with authentic circadian regulation of Srx.

Another question that remains to be answered relates to which of the Prxs is responsible for the observed diurnal and age-related changes in Prx-SO_2/3_ forms. The problem is that almost identical molecular weights of dPrx 1–3 (Figure 2 and calculated) might inhibit the ability to observe cycling of one of them with a generic reagent (Prx-SO_2/3_ antibody). There is still a possibility that all three dPrxs undergo rhythmic oxidation but with different circadian phases; this would also blunt effective amplitude unless assayed with reagents specific for the individual oxidized proteins. However, according to data obtained with mammalian Prxs [33], depletion of Srx by siRNA causes an approximately equal increase in SO_2/3_ forms of Prx3 and Prx1/2, which are distinguishable in mammalian and human orthologs by molecular weight, suggesting that Prxs can equally contribute to the formation of hyperoxidized Prx forms.

Finally, our studies have shown that the *srx* mutant does not recapitulate the phenotype of Prx1–4 knockdowns in terms of lifespan and resistance to oxidative stress caused by H_2_O_2_ consumption. If the observed phenotype of the *srx* mutant were due to a significant decrease in Prx activity due to oxidative inactivation, the characteristics of the individual Prx mutants would overlap, at least in part, with those of the *srx* mutant. However, underexpression of Srx and Prxs had different effects on resistance to oxidants and lifespan (Table 1).

There may be arguments for redundancy in the antioxidant function of Prxs, which is true for dPrx3, where only minor effects of underexpression on survival were observed ([12], Table 1). dPrx3 is a mitochondrial protein co-localized with another Prx, dPrx5 [12], and these Prxs can compensate for each other’s absence. However, both Prx1 and Prx2 are localized in the cytosol and do not appear to have functional overlap, since their underexpression has very different effects on physiology (Figure 5, Appendix A and Table 1). The phenotype of the *prx4* mutant, which is sensitive to oxidative stress [13], also differs from the phenotype of the *srx* mutant (Table 1). Since accumulation of catalytically inactive dPrx4 forms was not observed (Figure 2 and Appendix A), depletion of Srx activity could hardly affect the activity of this Prx.

In conclusion, catalytically inactive forms of hyperoxidized Prxs serve not only as a marker of changes in cellular redox and H_2_O_2_ fluxes, but can also play a role in an adaptive response, leading to a positive effect on the physiology of *Drosophila*.

## Figures and Tables

**Figure 1 antioxidants-10-00606-f001:**
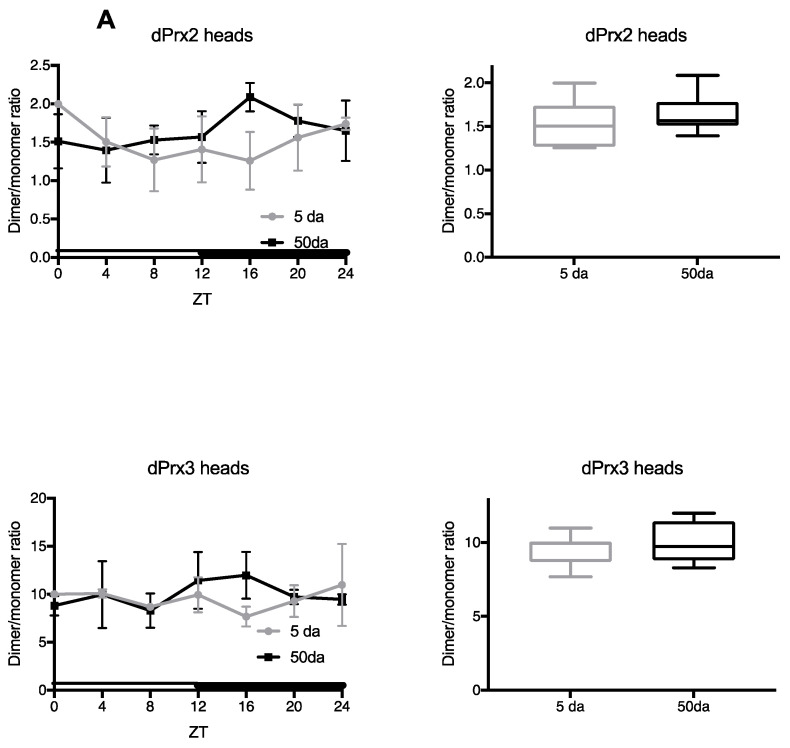
Analysis of the daily changes in the peroxiredoxin dimer to monomer ratio in young (5 da) and old (50 da) flies. Samples were prepared from the heads (**A**) and bodies (**B**) of Canton male flies maintained under light/dark (LD) conditions and collected for analysis at 4 h intervals. Extracts were made using a non-reducing lysis buffer, as specified in the Materials and Methods. Immunoblots were developed with antibodies specific for *Drosophila* peroxiredoxins 1–3. Ratios of signals corresponding to dimers and monomers are plotted on the Y axis. Zeitgeber time (ZT), where ZT0 is time of lights “on” and ZT12 is time of lights “off”, is plotted on the X axis. Data represent average values ± SEM obtained from three independent bio-replicates. Each replicate was normalized to the material collected from 5 da flies at ZT0 time point. Graphics on the right represent average 24 h dimer to monomer ratios in 5 and 50 da flies. Significance was analyzed by 2-way ANOVA with Bonferonni post-test. In the graphs shown on the left, # marks statistically significant differences between ZT0 and ZT4–24. In the graphs shown on the right, * marks a significant difference between 5 da and 50 da flies. Representative immunoblot images are shown in Appendix A.

**Figure 2 antioxidants-10-00606-f002:**
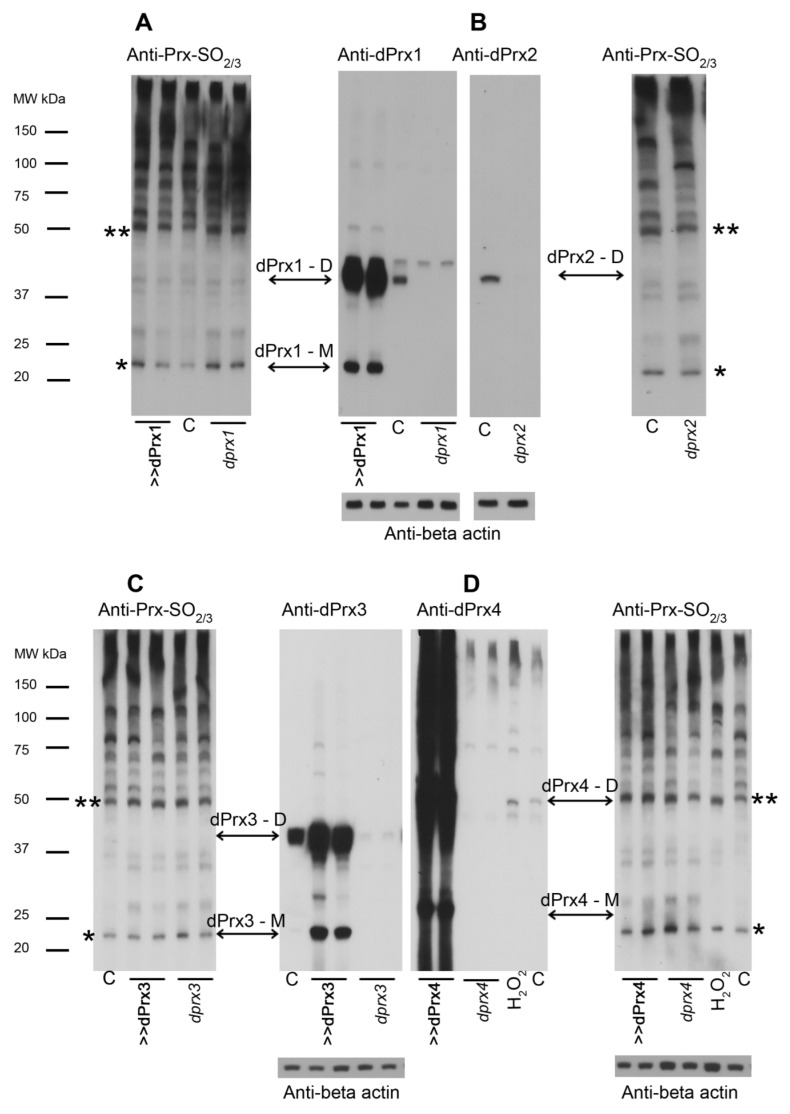
Analysis of peroxiredoxin hyperoxidation. Samples were prepared from control *yw* flies (**C**), control flies treated with hydrogen peroxide (H_2_O_2_), flies underexpressing dPrx1 (**A**), dPrx2 (**B**), dPrx3 (**C**) or dPrx4 (**D**) using corresponding UAS-RNAi transgenes crossed to Da-GAL4 driver (*dprx1, dprx2, dprx3 or dprx4*), and flies overexpressing dPrxs obtained by crossing UAS-dPrx transgenes to Da-GAL4 driver (>>dPrx1, >>dPrx2, >>dPrx3 and >>dPrx4). Extracts were made using a non-reducing lysis buffer containing alkylating agents (see Materials and Methods) to prevent artifactual oxidation of cysteines. Immunoblots were developed with the anti-Prx-SO_2/3_ antibody (Anti-PRX-SO_2/3_) and then stripped and re-probed with an antibody specific for *Drosophila* peroxiredoxins 1–4 (anti-dPrx1–4). Anti-actin antibodies (MP Biomedicals) reveal actin as a loading control. Arrows note positions of the proper dPrx dimers and monomers. Asterisks indicate positions of a putative Prx dimer (**) and monomer (*), which in the case of the dimer turned out to be non-specific bands recognized by the Prx-SO_2/3_ antibodies.

**Figure 3 antioxidants-10-00606-f003:**
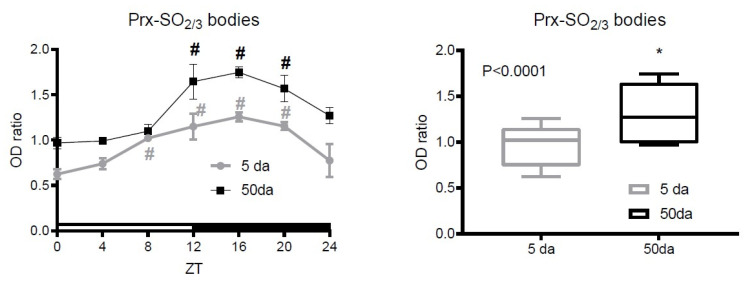
Analysis of daily changes in peroxiredoxin hyperoxidation in young (5 da) and old (50 da) flies. Canton flies were raised under LD conditions and collected for analysis at 4 h intervals. Samples were prepared from bodies of male flies. Immunoblots were performed under reducing conditions and developed with an antibody that recognizes Prx-SO2/3 forms and re-probed with specific Prx antibodies. In addition, Ponseau staining was used to standardize for loading. Signals obtained with anti-Prx-SO2/3 antibodies were normalized to signals obtained with Ponseau staining and ratios are plotted on the Y axis. Zeitgeber time (ZT), where ZT0 is time of lights “on” and ZT12 is time of lights “off”, is plotted on the X axis. Data represent average values ± SEM obtained from three independent bio-replicates. Significance was analyzed by 2-way ANOVA with Bonferonni post-test. Each replicate was normalized to the material collected from 5 da flies at ZT0 time point. Statistically significant differences are marked by asterisks. In the graphs shown on the left, # marks statistically significant differences between ZT0 and ZT4-24. In the graphs shown on the right, * marks a significant difference between 5 da and 50 da flies. Representative immunoblot images are shown in Appendix A.

**Figure 4 antioxidants-10-00606-f004:**
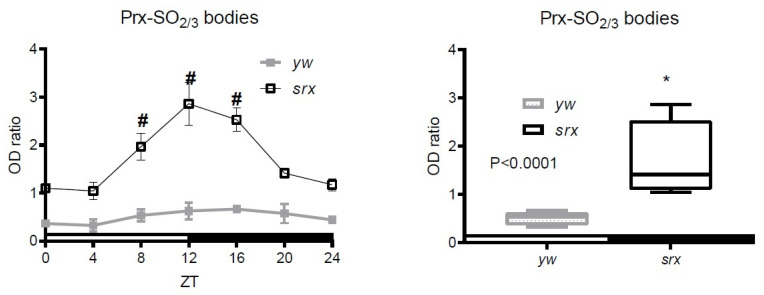
Analysis of of daily changes in peroxiredoxin hyperoxidation in the *srx* mutant. Samples were prepared from 5 da old control (*yw*) and *srx* mutant (srx). Flies were reared under LD conditions and collected at 4 h intervals. Immunoblots were performed under reducing conditions and developed with an antibody that recognizes Prx-SO_2/3_ forms and re-probed with anti-actin antibodies to control for loading. The results are mean ± SEM (*n* = 3). In the graphs shown on the left, # marks statistically significant differences between ZT0 and ZT4–24. In the graphs shown on the right, * marks a significant difference between 5 da and 50 da flies. Representative immunoblot images are shown in Appendix A.

**Figure 5 antioxidants-10-00606-f005:**
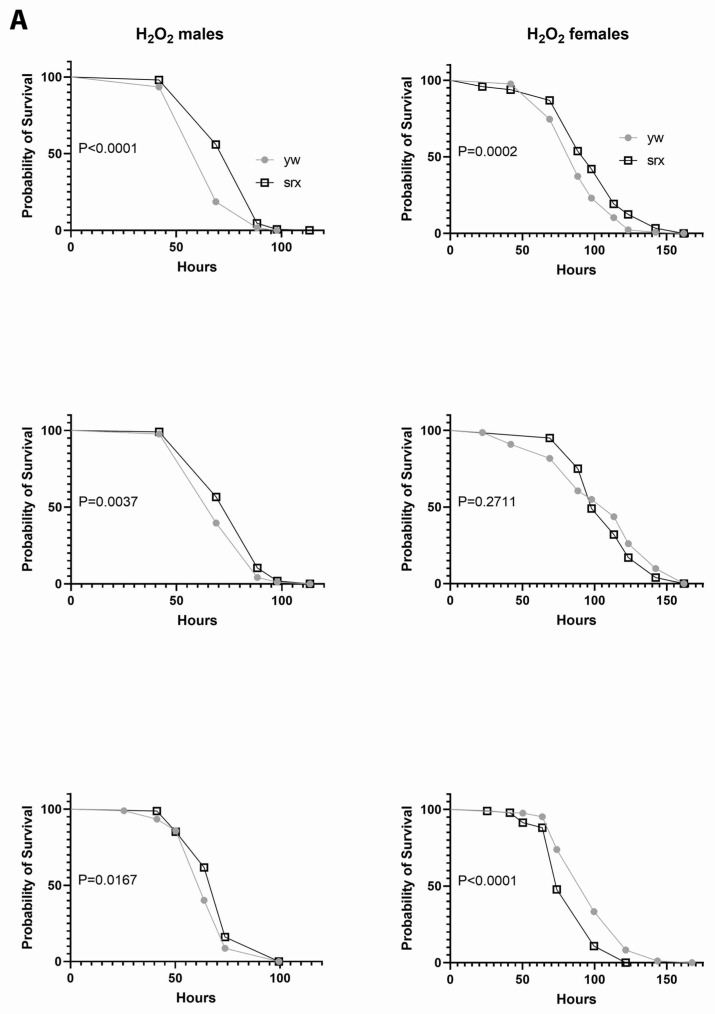
Effects of Srx underexpression on survival under normal and oxidative stress conditions. Control (*yw*) and the *srx* mutant (srx) flies were fed 1% sucrose solution containing 0.5 M H_2_O_2_ (**A**) or reared under normal conditions (**B**). Results were obtained with 3 different biological replicates. Approximately 100–150 flies for each fly line were used in the experiments. Statistically significant differences (*p* < 0.05) were determined by the log rank test and *p* values for each experiment are shown on the graphs. Representative immunoblot images are shown in Appendix A.

**Figure 6 antioxidants-10-00606-f006:**
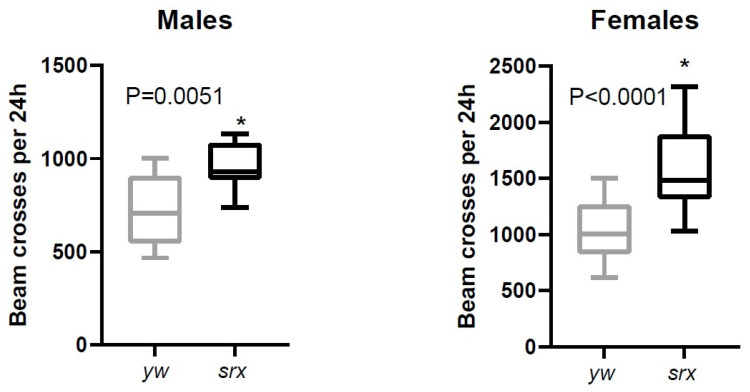
Effect of Srx underexpression on fly activity. The average activity of eight young (5 da) male and female flies was measured using the Trikinetics assay. The measurements were taken at 10 min intervals for five consecutive days. The data were obtained in two biological replicates. The average crosses of the beam per 24 h LD period are shown on the Y axis. Significance was analyzed by the *t*-test. Asterisks show statistically significant difference in activity between control (*yw*) and the *srx* mutant (srx). Representative activity data are shown in the graphs in Appendix A.

**Table 1 antioxidants-10-00606-t001:** Effects of underexpression of Prxs and Srx on lifespan and resistance to oxidative stress.

Under-Expressors	Resistance to Oxidative Stress	Lifespan
Prx1 (CG6888)	similar to control	similar to control
Prx2 (CG1633)	n/a	short lived, few progeny
Prx3 (CG5826)	slight decrease	similar to control
Prx4 (CG1274)	significant decrease	similar to control
Srx (CG6762)	similar to control or slight increase	similar to control or slight increase

## Data Availability

The raw data supporting the conclusions of this article will be made available by the authors, without undue reservation.

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
