# Peer review of "Hyperoxidation of Peroxiredoxins and Effects on Physiology of Drosophila"

_antioxidants, 2021, doi:10.3390/antiox10040606_

Round 1

Reviewer 1 Report

This paper by Radyuk and colleagues describes in detail the effects of increasing or decreasing levels of dPrx1, dPrx2 and dPrx3 in Drosophila, as well as decreasing Srx expression, on physiological functions and lifespan of the flies.  Resistance toward oxidative stress was tested by measuring survival after consumption of sucrose plus 0.5 M hydrogen peroxide, with a modest effect shown as a slight enhancement of survival with knockdown of the Srx expression.  Potential reasons for this finding are discussed, although more could have been done to address some of these (e.g., measurement of compensating dPrx5 level, Nrf2 activation).  Moreover, hyperoxidation levels and circadian cycling are evaluated according to the effects of aging and of Srx depletion, with rather expected results.  This is a really rigorous study with very interesting outcomes done in an important model organism.  However, I did have a few concerns elaborated below, particularly regarding the monomer-dimer ratios that apparently are determined simply from signal intensities, not corrected for the known differential recognition/signal that is obtained using many Prx-specific antibodies for oxidized versus reduced protein.

Major

Monomer-dimer ratios determined from band intensities of Western blots apparently assume that the oxidized and reduced forms of the proteins are recognized equally by the antibodies being used, but this is often not at all true, and this difference would have to be determined for each antibody-protein pair.  Typically the oxidized forms “light up” much more by polyclonal antibodies than do the monomeric, reduced forms.  This wouldn’t change the whole set of conclusions from these studies, but it is needed so that the ratios that are obtained are corrected for this and therefore more accurate.

On p. 9 at the top, dPrx3 underexpression is described as not having a pronounced phenotype, and one immediately wonders whether there is a Prx5-like protein expressed in mitochondria in these flies that is compensating.  This issue is then brought up in the last significant paragraph of the discussion, that “these Prxs can compensate for each other’s absence”.  But this protein has also been studied by this group before so it is not clear why the actual experiment was not done to determine whether or not “dPrx5” protein amount was increased in these flies.

Similarly in the discussion (middle of p. 13), there is discussion of Nrf2 activation differences potentially explaining some of the findings, but this is not measured here, missing an opportunity to test this hypothesis.

What is missing from the discussion is any reference to the relatively recent finding that there are a number of other substrates for Srx besides these typical 2-Cys Prx proteins (2018 paper from Yang and Carroll), which is very important here. Also another paper some time back (Tew’s group) suggested a deglutathionylating activity for Srx, although I am not certain that has been confirmed by any other group.

There is a problem with Fig. 5 in that the top lists “Figure 5A”, but then no B, and in fact there are three panels in total even though only two are described.  It is hard to tell what all these are.  Regarding the first sentence in Discussion, “even slightly extends flies lifespan” does not seem to be supported by data that shows significant differences, but this depends on understanding which panels are what in Fig. 5, since males show a slight difference in the middle panel and females in the bottom panel.  Perhaps the authors should reconsider/rephrase the sentence in the abstract saying that “the srx mutant was characterized by a higher physiological activity”.

The authors consider that they are unable to tell which of the three Prxs are undergoing hyperoxidation since their apparent MWs on gels are about the same (especially described on p. 14 paragraph starting on line 454), but couldn’t one antibody (say, the anti-Prx-SO2/3 antibody) be used to pull down hyperoxidized proteins, then samples blotted for each specific Prx (or vice versa)?

Minor

I imagine this is “Drosophila melanogaster”, but as far as I see only “Drosophila” is mentioned.

Line 40, because the 3 subgroups of 1-Cys, typical 2-Cys and atypical 2-Cys were introduced just above, when this says “2-Cys Prxs belonging to type 1-3”, which really means Prx1, Prx2 and Prx3, is a little confusing.  Otherwise, the description here about groups is accurate and very clear.

p. 3 line 135, it is misleading to just describe Prxs (actually, this should be “typical 2-Cys Prxs”) as alternating between monomeric and dimeric forms, since their oligomeric states in solution mostly alternate between dimers and (do)decamers. Perhaps this could be reworded to “Prx subunits alternate between monomeric and covalently-linked dimeric forms that migrate differently on non-reducing gels.” Just below that, line 139, “react with forming peroxides” is a little confusing and could just be reworded as “react with peroxides as they form”.

With Fig. 1, the right side shows values from 24 hr, but it is curious that, especially in some (dPrx3), the 24 hr point is pretty different from the 0 hr point even though it seems that they should be the same (back to end of dark and beginning of light period).  Was there some kind of treatment or difference at 0 hr that accounts for this?

p. 5 line 147, the beginning of the legend for Figure 1 lacks a title and starts in the middle of a sentence.

p. 7 line 208 in Fig. 2 legend, the phrase “a putative Prx dimer (**)” at 50 kDa is confusing to the reader, although explained more in the results text and especially in the discussion later as a non-specific band that might have been erroneously used for quantitation of hyperoxidized species in the circadian rhythm paper of Edgar et al. Perhaps in the figure legend call it “putative non-specific band”, and perhaps add “at a higher apparent molecular weight than most Prx dimers.”

p. 9 line 281 I suggest also including the motif around the Srx that is conserved, SFGGCHR. In Fig. S4 there is a problem with alignment in that line (probably need to replace dash with a hyphen or two). Related to this, also on line 281, the expression “high… homology” is not the best terminology, I suggest replacing “homology” with “identity” or “conservation”.  Below on that page, line 292, mentions inferring “that this protein does indeed possess sulfinic/sulfonic acid reductase activity…” but I would remove “sulfonic” since Srx proteins do not reverse that form.

p. 10, lines 305-306, insert “for” which fluctuations… and just below spell Sue Goo Rhee correctly. The srx mutant mentioned here is for what organism?

p. 13 line 368 mentions positive effects of the srx mutant on the survival of Drosophila under OS conditions… I strongly suggest adding “caused by H2O2 consumption” since that is only one type of OS, and not one that is very physiological. Same comment for p. 14 line 466.

p. 14 line 428, I suggest ending this with “but this was not rigorously addressed”.

p. 14 paragraph starting line 429, it is rather tricky to “catch” monomers fast enough with alkylation and not have artifacts where Prx dimers are formed during lysis, including from peroxides present in buffers (as elaborated by Winterbourn and Hampton papers), and it is hard to say for sure that Prxs are “usually” not monomers under normal conditions, but this is arguably the case from the literature. Also it is hard to equate dimer to monomer ratios to activity as it is more of a “flux” issue, affected by multiple things but especially substrate availability, whereas this is described more as a “static” ratio of oxidized and reduced forms. The point that oxidized (and especially hyperoxidized), if present as most of the population, is not available to immediately reduce peroxides is correct, though.  In fact, Prxs have to be actively cycling in the presence of high peroxide substrate concentrations in order to be significantly hyperoxidized, which is another point about considering flux rather than taking the more static view.  The presentation by the authors is reasonable, but perhaps could be improved a bit by mentioning “flux” in the cell.

There are multiple places where the word “data” is not used as a plural, so the verbs should be adjusted.

Typos or grammar problems:  p. 2 line 47 “Drosophla” (also not italicized); p. 3 line 127 “to” should be “like”; line 132, I suggest changing “and” to “and/or” regarding reductants;  p. 8 line 232 “for a loading” and line 243 “deecrease”; p. 13 lines 370-371, correct to beneficial effects “on” Prxs… and with gain “of” chaperone ; p. 14 lines 419-420, something is wrong with the reference numbers and “repots”.

Author Response

Dear Publishers,

I am very grateful to the Reviewers and Editors for reviewing the manuscript. Below, I provide detailed responses to each question in the Specific Comments. All suggestions for improving the manuscript have been addressed and all minor issues raised by Reviewer 1 and 2 were addressed.

Reviewer 2 Report

In the paper by McGinnis et al “Hyperoxidation of Peroxiredoxins and Effects on Physiology of Drosophila” authors elucidated all four 2Cys peroxiredoxins. They looked at the daily fluctuation of the redox state of the proteins in young and old flies and how it correlated with locomotor activity as an indicator of physiological state. Also, the flies were exposed to oxidative stress via including hydrogen peroxide in feeding substrate. Genetic manipulations were used to have varying level of peroxiredoxins and sulfiredoxin. Overall it is wide and well executed research which resulted in interesting findings. Authors unambiguously demonstrated that accumulation of hyperoxidized peroxiredoxin in Drosophila resulted in increased locomotor activity. Moreover, survivorship was not diminished in flies with downregulated sulfiredoxin neither in control or under oxidative stress conditions. The paper will be of interest to the redox community.

Minor issues.

  1. The beginning of the Fig.1 capture has been lost.
  2. Fig.2. Put the labelling of pictures into the capture.
  3. Table 1. It would be good to have it structured in real table for easy reading.
  4. Fig.5.  B is absent.
  5. Fig.6S. It wouldn’t hurt to label time points immediately under lanes on the picture.
  6. Abbreviations of Prx and Prdx for peroxiredoxins have been used but recently have been rationalized to Prdx.

Author Response

(The authors gave the same response as above.)

Round 2

Reviewer 1 Report

The responses by the authors are largely acceptable, except for one major and one minor issue.

First, the authors misunderstood the comment about dimers and monomers of Prxs not generally being recognized with the same affinity/efficacy by anti-Prx antibodies (in our experience).  They took this to be referring to the antibody recognizing the hyperoxidized forms, which was not the point being made.  Also they referred in this response document to the monomers and dimers as both being “reduced”, which is odd since the dimers indicate disulfide bond formation.  The point is that a given antibody often (in our hands) does not recognize these upper (dimer) and lower (monomer) bands equally, so simply deriving ratios from the intensities of the bands may be skewing the true ratios if not corrected for this.  It would not be difficult to load samples of known (or same) concentrations in fully dimeric or fully monomeric form (best to leave an empty lane between, if reductant is included in some of the samples) and test for those intensities for each of the Prxs and antibodies being studied/used.  If they are identical, then that issue is addressed.  If not, then the TRUE ratios would be calculated using the appropriate correction factor.  Unless the gels or membranes are treated with reductant AFTER running the gel to resolve dimers and monomers but BEFORE the antibody is incubated with the membrane, there is a potential (and common) problem of having the two forms be recognized differentially by the antibodies applied.  I didn’t not see an indication that this was how the experiments were done.

Second, p. 2 line 66 now mentions Jafrac1 as “more homologous”, which is still the problem I was hoping to see addressed.  This can be replaced with “more closely related” (they ARE homologues, or “closely related” homologues).

Author Response

Response to Reviewer #1 comments:

Point 1: First, the authors misunderstood the comment about dimers and monomers of Prxs not generally being recognized with the same affinity/efficacy by anti-Prx antibodies (in our experience).  They took this to be referring to the antibody recognizing the hyperoxidized forms, which was not the point being made.  Also they referred in this response document to the monomers and dimers as both being “reduced”, which is odd since the dimers indicate disulfide bond formation.  The point is that a given antibody often (in our hands) does not recognize these upper (dimer) and lower (monomer) bands equally, so simply deriving ratios from the intensities of the bands may be skewing the true ratios if not corrected for this.  It would not be difficult to load samples of known (or same) concentrations in fully dimeric or fully monomeric form (best to leave an empty lane between, if reductant is included in some of the samples) and test for those intensities for each of the Prxs and antibodies being studied/used.  If they are identical, then that issue is addressed.  If not, then the TRUE ratios would be calculated using the appropriate correction factor.  Unless the gels or membranes are treated with reductant AFTER running the gel to resolve dimers and monomers but BEFORE the antibody is incubated with the membrane, there is a potential (and common) problem of having the two forms be recognized differentially by the antibodies applied.  I didn’t not see an indication that this was how the experiments were done.

RESPONSE 1: We are grateful to the reviewer for finding the confusing term "reduced" when referring to Prx dimers and monomers. In fact, we meant that the dimers and monomers do not contain overoxidized cysteines. We also agree with the reviewer that polyclonal antibodies can have different reactivities with Prx monomers and dimers. Moreover, the bands obtained under reducing conditions are usually sharper and more intense, probably due to the better accessibility to various epitopes, as we observed in the experiment shown in Figure S2C, which has now been added to Supplementary Material for greater clarity and additional support of our findings. However, our goal was not to determine the absolute concentrations of various forms of Prxs, but to determine the relative diurnal and age-related variations in the proportions of different forms of Prxs. Since the experiments were carried out by placing all the material in the same gels, we were indeed able to analyze the differences in temporal changes in the dimer / monomer ratio in different samples.

Point 2: Second, p. 2 line 66 now mentions Jafrac1 as “more homologous”, which is still the problem I was hoping to see addressed.  This can be replaced with “more closely related” (they ARE homologues, or “closely related” homologues).

RESPONSE 2: We changed the text “…more homologous…” to “…more closely related…”, as suggested by the reviewer (line 66).